# On the Playing Field to Improve: A Goal for Autism

**DOI:** 10.3390/medicina56110585

**Published:** 2020-10-30

**Authors:** Luigi Vetri, Michele Roccella

**Affiliations:** 1Department of Sciences for Health Promotion and Mother and Child Care “G. D’Alessandro”, University of Palermo, 90127 Palermo, Italy; 2Department of Psychology, Educational Science and Human Movement, University of Palermo, 90128 Palermo, Italy; michele.roccella@unipa.it

**Keywords:** autism, ASD, sport, soccer, football

## Abstract

In recent years, there has been a renewed attention to lifestyle-based interventions in people with autism spectrum disorder. The positive effects of physical exercise programs have been well documented both in healthy people and in people with disabilities in the fields of psychological well-being, cognitive outcome and medical health. There is much less evidence about the opportunity to attempt a team-group sport for people with autism. Although researchers seem to suggest an overall positive effect, playing team sports for people with autism spectrum disorder (ASD) means dealing with difficulties in social interactions and limitations in motor functions. This narrative review aims to report studies about the effects, improvements and difficulties that people with autism have to face when they play the world’s most popular team sport: soccer.

## 1. Introduction

When, in the dawn of time, perhaps in ancient China or in the ancient land of the rising sun, a group of men began to delight in pushing a spherical object towards a fixed point, overcoming the opposition of another group that had to prevent it, no one had thought that such a game in the distant future could become an opportunity for development for autistic subjects unable to control the afinalistic repetition of some of their gestures and some of their actions.

Over time, that playing field could have changed into a therapeutic and rehabilitative space, and that pastime game could have become an ecological, significant activity aimed at the maturation of neurodevelopmental skills in subjects with an extreme poverty of interests and significant difficulties of neurodevelopment, social interaction and communication abilities.

Playing the ball together with others is not simply kicking a ball. Rather, it is a complex activity at a neuropsychological and neuro-anatomical functional level that requires an extremely diversified “motor vocabulary”.

Movements must overcome their mechanical dimension, they must coordinate with each other and become finalized gestures, increasingly economical and, in a more advanced phase, also strategic and elegant. Gestures, in turn, must be planned, modulated and controlled not only in relation to the action program, but also to other numerous conditions, such as the rules of the game, the presence of teammates and rivals, managing the relational dimension, stress and physical contact.

Playing football, therefore, means activating a set of “frontal” and “fronto-limbic” structures and neuropsychological competences involving self-regulation, concentration and attentional maintenance, divided attention, working memory, flexibility, problem solving and relevant emotional control [1,2].

According to an extensive clinical literature, autism spectrum disorder (ASD) may be viewed as a complex neurodevelopmental disorder characterized by difficulties in social interaction and communication, restricted and repetitive interests and behavioral patterns and impaired sensory processing [3], due to an intricate, partially unknown, multifactorial etiology. It can display a great phenotypic heterogeneity, expressing in a broad constellation of linguistic, personality, socio-behavioral traits associated with several neurodevelopmental, medical and psychiatric comorbidities that further complicate such complex picture. Such diversified phenomenology has supported a plethora of therapeutic and rehabilitative approaches that often confuse the caregivers and increase parental stress [4,5,6,7,8].

The prevalence of autism has been constantly increasing over the last decade from 2–5/10,000 to 1:59 children (1 in 37 boys and 1 in 151 girls). The frequency in males is four times greater than females [9].

The positive effects of exercise in healthy people have been well documented in literature, especially in psychological well-being, cognitive outcome and medical health [10,11].

Earlier studies and reviews have underlined the positive benefits of physical activity and exercise interventions on individuals with ASD [12,13,14,15,16,17] with regard to reduction of motor deficits [18], obesity and overweight [19], and to improvements in cognitive, academic [20,21], behavioral and socio-emotional skills [22,23,24]. Although there is much literary evidence about health benefits of physical exercise programs, much less are studies about the opportunity to practice a sport for people with ASD. Practicing a sport for people with ASD means dealing with two main barriers: difficulties in social interactions, that can express as tactile defensiveness resulting from an abnormal response to sensory stimulation [25], and limits in motor functions. People with ASD compared to individuals with typical development have more difficulties in balance, gait, movement speed, motor control and joint flexibility [26,27].

The most frequent physical exercise programs in people with ASD include swimming, jogging, walking, horseback riding, cycling and weight training, martial arts, yoga, dancing. These activities are mostly individual exercises, and when they are performed in groups (e.g., martial arts, yoga, dancing), social interactions are limited, and they do not constitute the unavoidable foundation like in team sports.

Although individual exercise interventions have positive benefits in people with ASD, such as more personalized approaches to every patient [28] and less stress and negative feelings that can develop toward group activities due to social impairment [29], team sports may have undeniable advantages, they mainly increase social attitudes and communication skills [30].

Soccer is the world’s most popular team sport. Its relative safety, the cheap equipment, the environmental accessibility and intuitive aim and the skills required are probably the causes of soccer success. Accordingly, it is not surprising that soccer is more practiced among people with intellectual disability, and new programs and courses for people with special needs are constantly being developed [31].

In the literature, there are very few studies about the impact of team sporting programs such as soccer in people with ASD.

Therefore, the purpose of the present narrative review is to examine the effects on individuals with autism when participating in a soccer-training program and to understand if a soccer program can be proposed as a model of innovative and effective ecological rehabilitation.

## 2. Aims and Methods

The aim of the current non-systematic review is to analyze the research results found in the literature in terms of improvements and difficulties encountered by people with ASD who decide to take part in a soccer program.

Thanks to the main research findings, this narrative review helps to shed some light on the role that a soccer training program can play within a rehabilitation path.

To this end, several articles, published over the years, were reviewed by performing a search using the following syntax “autism” [Title/Abstract] OR “Asperger” [Title/Abstract] OR “pervasive developmental disorders” [Title/Abstract] AND “soccer” [Title/Abstract] OR “sport” [Title/Abstract]. References were identified through electronic database searching in CENTRAL, Ovid MEDLINE, Embase.

The final database search was run on July 2020.

## 3. Motor Impairment in ASD

The two core features of ASD are an impairment in social communication and interaction and an atypical pattern of repetitive behaviors [3]. However, numerous pieces of evidence in the literature suggest that children with ASD also have several motor impairments related to the overall development [32]. Green et al., using the Movement Assessment Battery for Children (M-ABC) in a large group of children with ASD (*n* = 101:89 males, 12 females; mean age 11 years), found that 79% of children had definite movement impairments and a further 10% had borderline problems. Interestingly, children with childhood autism and an IQ less than 70 were more impaired than children with broader ASD and higher IQ, underlining a severer neurological impairment that contributes to both motor impairment and intellectual disability and social skills [33].

These results are consistent with those of other studies that found positive statistically significant correlations between overall gross motor and social skills [34,35,36,37].

Less consistent is the relationship between gross motor functions and severity of ASD. Fine motor and gross motor skills significantly predicted autism severity according to the results by MacDonald et al. [38], while other studies found that gross motor ability was not related to ASD severity [39,40].

Motor impairments in people with ASD may be suspected very early even during the first year of life [41], and they can be observed both during infancy [42,43] and into adulthood [44].

The motor deficits usually observed in children with ASD are fine and gross motor delays [45], postural control as a result of anomalies in vision and proprioception information processing [46], joint flexibility [47], unstable balance [48], gait cycle abnormalities [49], motor coordination deficits [50], less accurate manual dexterity [51] and hypotonia [52]. Moreover, children with ASD showed worse ball skills compared to healthy children [53,54].

## 4. Physical Exercise and ASD

Six reviews analyzed the effects of a physical exercise program on people with ASD.

Petrus et al. in their systematic review synthesize the evidence from studies examining the effects of physical exercise on stereotypic behaviors in children with ASD. They found seven articles that met the inclusion criteria and concluded that physical exercise produces short-term improvements in stereotypic behaviors, especially when practicing high-intensity training. However, the authors observed that the studies analyzed had a high heterogeneity of research designs and types and intensities of interventions, making it difficult to determine a specific better exercise for children with ASD.

A similar more recent review with meta-analysis by Ferreira et al. analyzed eight studies including 129 children with an average age of about 9 years. The authors found a significant reduction in the number of occurrences of stereotypical behaviors after physical exercise programs but highlighted that the studies analyzed did not establish which type of exercise intervention programs was more effective and which were the effects of physical exercise programs on children under 8 years old [13].

Another systematic review by Lang et al. analyzed 18 studies concluding that physical exercise programs produce improvements in behavior (e.g., reduced stereotypy, aggression or self-injury), academics performances (e.g., increased amount of time on-task or increased accuracy in academic responding), physical fitness (e.g., increased endurance or strength), or increased exercise behavior (e.g., more time engaged in exercise). The most frequent improvements were reduced stereotypy or self-stimulatory behavior [14].

Similar results were obtained by Bremer et al. who systematically reviewed evidence in the literature regarding the behavioral outcomes of exercise interventions in individuals with ASD aged ≤ 16 years. The authors found improvements in stereotypic behaviors, social-emotional functioning, cognition and attention—in particular, interventions involving horseback riding and martial arts seemed to produce the greatest results [15].

A meta-analysis by Sowa and Meulenbroek evaluated 16 studies reporting a total of 133 children and adults with ASD. The authors found an overall improvement score in social and motor deficiencies in 37.5% of patients with ASD. Interestingly, children with ASD profited more from individual interventions (48.57%) rather than from group interventions (31.54%). Surprisingly, in social interaction domain, the individual interventions also generated larger positive effects (71.43%) than the group interventions (26.37%) [17].

However, not all studies support the benefit effects of physical exercises, especially when analyzing the core symptoms of ASD. Howells et al. in their recent review and meta-analysis analyzed the effects of group-based organized physical activity on social and communicative outcomes in children with ASD. Their statistical results showed a non-significant effect for communication outcomes and a significant small–medium improvement in the overall social functioning. However, the authors emphasized that group-based organized physical activity programs are often heterogeneous, and, therefore, it is very difficult to draw general conclusions [55].

## 5. The Neuropsychological Benefits of Soccer in Typically Developing Individuals

The benefits of regular exercise and of practicing a sport on physical and psychological health are undeniable. Physical activity decreases the risk of hypertension, obesity and overweight, cardiovascular disease, type 2 diabetes, stroke and developing dementia [56].

Playing a sport in children also has several psychological and social health benefits such as improved self-esteem, social interaction, life satisfaction and fewer anxious/depressive symptoms and social isolation [57]. Moreover, longitudinal physical activity programs have positive effects on children’s executive functions, attention and academic performances [58]. The benefits of team-sport programs go beyond improvements simply related to physical activity due to the social nature of these sports. Eime et al. in their systematic review proposed a socio-ecological model of health through sport. It includes two dimensions of sport participation, namely, individual–team and informal–organized, and three types of health outcomes, namely, physical, psychological and social. Only team sport strongly contributes to psychological and social outcomes, while the individual forms contribute less to psychological outcomes and very little to social outcomes [57]. This is in accordance with the notion that a team-sport program could represent a model of ecological rehabilitation in people with impaired social behaviors such as in ASD.

Few studies analyze the psychological effects and the social outcomes of a specific sport, and, traditionally, researchers consider together all individuals practicing a team sport without distinctions about the type of team sport. However, there are some exceptions in the literature. For instance, Huijgen et al. in their study administered neuropsychological tests to 47 elite youth soccer players and 41 sub-elite youth soccer players (aged 13–17 years) in order to evaluate cognitive functions, measuring working memory, inhibitory control, cognitive flexibility and metacognition.

Elite youth soccer players showed better inhibitory control, cognitive flexibility and especially metacognition than sub-elite youth soccer players. No differences were detected in the “lower-level” cognitive processes such as reaction time and visuo-perceptual abilities [59].

A similar cross-sectional study on a sample of 131 male children aged 10–12 years, 70 non-athletes and 61 soccer players showed that practicing soccer regularly during preadolescence determines faster responses and better executive control (reduced interference from distractors) [60].

Other studies used standardized neuropsychological assessment tools, assessing executive functions in soccer players with similar results. Professional soccer players outperformed amateur soccer players in executive function tests, and both groups performed better than the general population [61].

However, there are two important elements to consider. First, the existence of possible selection bias in the sport context; that is, individuals with better cognitive functioning can choose more frequently active lifestyles. Second, the only longitudinal study, to our knowledge, assessing executive functions in 304 high-performing male youth soccer players (10–21 years old) revealed that the developmental trajectories of executive functions across the lifespan are the same between individuals with long-term exposure to soccer-specific training and the general population [62].

Making conclusions about the neuropsychological consequences of a soccer training program due to the lack of studies is very difficult. Further studies are needed in order to elucidate if and what training is effective to provide improvements in cognitive and socio-emotional skills. However, the improvements underlined by the above-mentioned studies in social skills and in executive functions, often impaired in people with ASD showing difficulties in information processing, planning, and self-regulating emotions, make soccer a promising and attractive model of ecological rehabilitation.

## 6. ASD and Soccer

### 6.1. Children

All literature reports (to the best of our knowledge, only four studies) about soccer programs for children with ASD are very recent, and they are summarized in Table 1.

Hayward et al. (2016) in their study aimed to verify the efficacy of a 6-week, community-based, adaptive soccer program for a small group of 18 children with ASD and a mean age of 9.77 (SD = 2.15). The authors evaluated physical activity outcomes such as pre- and post-soccer skills, participant attendance and parent satisfaction. The aim of their soccer program was to teach children with ASD basic soccer skills, giving them the opportunity to have fun and interact with peers. There were six sessions (once a week) that lasted 90 min. Each session consisted of soccer skill training, several opportunities for social skill development and a part for collecting physical activity parameters and developments. Each child during sessions was assisted by one or more “buddies” depending on his/her autism severity. Buddies were selected by volunteer networks, and some of them were doctors of physical therapy certified in applied behavioral analysis (ABA). ABA strategies were used to improve children’s ability to stay on task, sometimes using tokens and picture exchange communication systems (PECS).

Their results supported the feasibility and effectiveness of a soccer program because they found improvements in kicking accuracy (*p* = 0.048) and 15-yard agility time (*p* < 0.001). Parent overall satisfaction was very good, 100% recommended it to a friend, and in their opinion, their children were more active and enjoyed playing soccer [63].

‘Calcio Insieme’ (Soccer Together) is an extremely interesting project promoted by Roma Cares Foundation (a non-profit organization linked to AS ROMA and to A.S.D. Accademia di Calcio Integrato) aimed at improving education and culture together with sporting values through soccer. In the Soccer Together program, Cei et al. (2017) recruited 30 children with ASD (6–13 years old, 27 boys and 3 girls) to study the effects of a training program based on teaching soccer.

All children were subjected to the initial and final quantitative motor assessment. Differently, the authors used a qualitative approach to evaluate the psychosocial skills at the beginning and at the end of the training period through interviews with their parents and schoolteachers. The results demonstrated that parents and schoolteachers perceived that most children with ASD improved their psychosocial and communicative abilities.

The quantitatively assessed motor skills showed a significantly improvement in the six following tests out of ten: walking between the cones, running between the cones, roll on the mat, high jump (three obstacles 20/30 cm), grab (five launches from 1 to 5 m away from the instructor) and stay balanced on jellyfish [64].

Chambers and Radley in their study used a different approach. They chose a peer-mediated intervention to promote skill acquisition in children with ASD. The authors selected three male students with autism (respectively 12, 12 and 11 years old) and instructed a common 14-year-old female peer interventionist for all three participants. The soccer skills evaluated were throwing, kicking and defending. The study had a multiple probe design. In the sessions with a three-time-per-week frequency, the peer explained and demonstrated the soccer skills to children with ASD and after watching the child’s attempt, the peer would correct any mistakes. Each child had 10 opportunities for each skill to demonstrate a good skill accuracy. The study was composed by three phases: baseline, intervention and maintenance. Phase changes were made in relation to participants’ skill accuracy. At the end of the study, all three participants rapidly acquired the targeted soccer skills; moreover, skill accuracy persisted over time in the absence of any interventions [65].

### 6.2. Adults

A similar project named Game of Life (GOL) is a soccer program promoted by “Mifalot Chinuch” (Educational Operations in English), a non-profit organization with the aim to improve the quality of life of children or adults with social difficulties or disabilities. Nineteen adults with ASD (mean age 32 SD = 8.00; range: 19–55 years; 16 males, 3 females) were recruited from residential care centers, and all study participants were added to a different soccer team. This is the only analyzed study using a comparison group to evaluate its results. The autism group was compared to a group of 50 adults with intellectual disability. The GOL program consists of 6 months of one weekly session and a final friendly tournament. A typical session lasts about 90 min and includes the following consecutive phases: (a) educational briefing, (b) warm-up, (c) fitness training, (d) technical and tactical training, (e) a game and (f) debriefing. All participants underwent a complete battery of tests assessing soccer skills, physical fitness skills and mobility skills at the beginning and at the end of the soccer program. Strangely, people with ASD showed no significant improvements in total soccer skills, while presenting a significant improvement in physical fitness skills (Sit and Reach Test) and in mobility skills (reduction in Timed Up and Go test). However, the intellectual disability group showed better improvements than the ASD group, which, in the authors’ opinion, is probably because the GOL program needs to be improved to become more suitable for people with ASD [66].

## 7. Limitations and Future Directions

The main limitation of this narrative review is the small amount of studies analyzed. The lack of literature reports makes it difficult to draw conclusions on the impact of soccer programs on people with ASD. Moreover, the analyzed studies have different settings, group compositions, soccer programs, often lack of control group and use different measures to evaluate the outcome results.

In order to draw overall conclusions, we hope that the number of studies about the therapeutic role of a team-sport program will increase in the future and that these studies will use a more rigorous and shared protocol including, for instance, an inactive control group or a control group having a different rehabilitation approach. In our opinion, a more intensive training program can also produce positive and better effects on people with ASD.

In the end, we hope that greater attention will be given to the interpretation of results, not focusing only on motor improvements, but rather on core symptoms of autism, especially on social interaction. Moreover, it is essential to use standardized tools to make these evaluations.

## 8. Conclusions

In summary, the current narrative review reports the overall positive indications of soccer programs for children with ASD. The well-known positive link between physical activities and well-being, especially in a social context, may represent an excellent and additional healthy outlet against psychiatric disorders. Therefore, it is not surprising that more and more families choose a lifestyle-based intervention in conjunction with clinic-based interventions for their sons with disabilities.

Families of children with ASD often encounter a number of difficulties to participate in a soccer program. However, the aforementioned studies show that parent-reported ratings indicate an overall excellent satisfaction after a team-sport program. Moreover, many children with autism in countries where soccer is very popular, like all children, would like to learn to play soccer, and their passion can be put to good use to reduce their social anxiety. Soccer-related programs have also shown multifaceted potential benefits in behavior, social competence, developmental outcomes, physical and motor skills.

Due to this new attention to lifestyle-based and team-sports interventions, we hope that further studies with a more solid methodology and minor heterogeneity of protocols and sessions could shed some light on the impact of team sports on the social behavior and global outcomes of people with autism.

The idea that “football is medicine” and that it can produce positive long-term physical and psycho-social training-induced effects is not new [67]. However, in our opinion, before proposing a soccer program as a model of ecological rehabilitation, more robust evidence is needed to clearly define the optimal protocol of the training program. At present, it is very difficult to establish an optimal training regimen for people with ASD; the literature gives fragmentary evidence about this point that seems to indicate that an inclusive approach, which enables peers to interact and cooperate with each other, supported by properly trained assistants in sessions with appropriate length and frequency, could be a good starting point. We also hope that in the near future, a prescription of a team-sport program could find a place alongside current clinical and rehabilitative interventions and also that people with ASD could experience the joy of practicing the most beautiful sport ever played.

## Figures and Tables

**Table 1 medicina-56-00585-t001:** Soccer programs for children with autism spectrum disorder (ASD) (F/M: Females/males; SD: Standard deviation).

Study	Sample Size	Sex F/M	Mean Age in Years, (SD); [Range]	Duration	Soccer Program	Outcome	Outcome Measures
Hayward et al. 2016 [63]	18	5/13	9.77 (2.15)	6 weeks	adaptive soccer program	kicking accuracy and 15-yard agility time	U.S. Youth Soccer Training Activities Manual
Cei et al. 2017 [64]	30	3/27	[6–13]	5 months	training program	psychosocial and motor skills	qualitative and quantitative measures
Chambers and Radley 2019 [65]	3	0/3	12 (0.5) [11,12]	9 weeks	peer-mediated intervention	accuracy of soccer skills	quantitative measures
Barak et al. 2019 [66]	19	3/16	32 (8.00) [19–55]	6 months	multiphase sessions	physical fitness and mobility skills	Football Athlete Skills Assessment

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
