# Peer review of "On the Playing Field to Improve: A Goal for Autism"

_medicina, 2020, doi:10.3390/medicina56110585_

Round 1
Reviewer 1 Report
The authors present a narrative review of the effects of exercise on people with autism spectrum disorder (ASD) culminating in a description of four studies of the effects of training soccer on people with ASD.
This is an important topic that likely merits publication with revisions.
The manuscript can be improved by describing demographic details (age mean +/- standard deviation and range) and sex for descriptions of individual studies.
The manuscript can be improved by adding a section on limitations.
Indicate that this was not a systematic review. Indicate that several commonly used databases (PubMed, ScienceDirect, and Web of Science) were not included.
What additional measure will improve future studies?
Based on their review of the studies, the authors could help readers by proposing an optimal training regimen for people with ASD.
Page 3
Line 111
State the exact number in place of "a large number. . . ."
Line 132
Delete "At least. . . " at the beginning of the line. If you identified more then include them.
Page 4
Section 5. Indicate in the heading that this section treats with individuals with typical development, not ASD.
Page 5
Section 6
Include separate subheadings for children and adults
Line 214
In addition to the mean, include the standard deviation and the range.
Line 245
Include the range of ages.
Page 6
Lines 255-6 What were the sexes of the students?
Author Response
Dear Reviewers,
I would like to thank the reviewers for your valued comments and suggestions to the article. As you requested, we made all the necessary changes in our manuscript to address the reviewers’ concerns and we detailed below how the points raised by the referees have been accommodated. The main changes are written in red in the text of the manuscript as well as the answers point-by-point below. From the changes made in the revised manuscript and responses provided below, I hope you are convinced that we have adequately addressed the reviewer’s concerns and made the paper better. If there are any further questions, please feel free to let me know.
The authors present a narrative review of the effects of exercise on people with autism spectrum disorder (ASD) culminating in a description of four studies of the effects of training soccer on people with ASD.
This is an important topic that likely merits publication with revisions.
The manuscript can be improved by describing demographic details (age mean +/- standard deviation and range) and sex for descriptions of individual studies.
Thanks for the suggestion. The data requested have been added in table 1. Unfortunately, not all demographic details (age mean +/- standard deviation and range) are available in the four studies analyzed.
The manuscript can be improved by adding a section on limitations.
Thanks for the suggestion. A new section entitled “Limitations and future directions” was added (please see lines 280-292).
Indicate that this was not a systematic review. Indicate that several commonly used databases (PubMed, ScienceDirect, and Web of Science) were not included.
We underlined that this is a narrative review (please see for instance lines 22, 88, 96...). We better explained that this is not a systematic review in the methods (line 96). Moreover, you can find the databases used in lines 101-102.
What additional measure will improve future studies?
Thanks for the suggestion. We discussed this point in the new section added “Limitations and future directions”.
Based on their review of the studies, the authors could help readers by proposing an optimal training regimen for people with ASD.
Many thanks for the suggestion. I think that this is a crucial point of the topic of my paper. Unfortunately, the studies analyzed at present are very heterogenous and it is very difficult to delineate which is the best training program for people with ASD. We better discussed this point in the conclusion (please see lines 314-318).
Page 3
Line 111
State the exact number in place of "a large number. . . ."
We specified the exact number of individuals with ASD in line 110 as requested.
Line 132
Delete "At least. . . " at the beginning of the line. If you identified more then include them.
We deleted “At least” as suggested.
Page 4
Section 5. Indicate in the heading that this section treats with individuals with typical development, not ASD.
Thanks for the suggestion. I corrected as you indicated.
Page 5
Section 6
Include separate subheadings for children and adults
I added separate subheading for children and adults, thanks for the suggestion.
Line 214
In addition to the mean, include the standard deviation and the range.
I added the SD, the range is not available.
Line 245
Include the range of ages.
I added it.
Page 6
Lines 255-6 What were the sexes of the students?
They are males (line 248).
Reviewer 2 Report
Thank you for inviting me to review this paper entitled “On the playing field to improve: a goal for autism”. The authors aimed to analyze the research results found in literature in terms of improvements and difficulties encountered by the people with ASD who decide to take part in a soccer program.
I have really appreciated the authors’ efforts and acknowledge the novelty and the interest of the topic. However, I have some suggestions that the authors may take into account to improve the quality of their manuscript.
First, even if this is presented as a narrative review, it seems more a hybrid between a narrative and a systematic review. In fact, a search string was used and the results of the search have been reported in a systematic way, with a table and so on. The authors should decide if they want to keep the “narrative” or the “systematic” modality. In this second case, it would be better to implement the Methods section by adding more information of interest for the authors, such as the inclusion criteria for the studies (Participants, Intervention, Outcome, Study design), the timeframe of the included studies, the methodology about data extraction of results report, and so on….
In case you decide to keep it as a narrative review, I would suggest giving some critical interpretation of the results, instead of simply describe the study that deals with a topic or another. For instance, in the paragraph about “The neuropsychological benefits of soccer”, the authors simply report the neuropsychological effects of football in players, but do not explain the rationale for soccer being beneficial for ASD. The same for the other paragraphs. This is important to lead the reader to the “core” paragraph of the review (paragraph 6).
Also, in the Table, I suggest to insert more specific information, e.g. mean age/range, sex of participants, type of treatment, treatment duration, comparison (if there is any), outcome (not improvements…), outcome measures (if there are validated scales). This would make the Table clearer for the reader.
In general, I suggest the authors be more critical (in sense of data interpretation) and gives also a suggestion which might be useful for clinical practice/rehabilitative programs.
Minor comments:
- Line 57-59: “ASD includes a constellation of language, personality, social-behavioral phenotypic traits 58 combined with several frequent neurodevelopmental, medical and psychiatric comorbidities that 59 further complicate the complex picture of autism manifestations.”: Please, delete. It is a repetition of a previous sentence.
- Page 2: I suggest to move the aim of the study to the end of the Introduction, instead of merging it with the Methods section, which should be improved, as reported above.
- Line 109: [3].3 --> [33].
Author Response
Dear Reviewers,
I would like to thank the reviewers for your valued comments and suggestions to the article. As you requested, we made all the necessary changes in our manuscript to address the reviewers’ concerns and we detailed below how the points raised by the referees have been accommodated. The main changes are written in red in the text of the manuscript as well as the answers point-by-point below. From the changes made in the revised manuscript and responses provided below, I hope you are convinced that we have adequately addressed the reviewer’s concerns and made the paper better. If there are any further questions, please feel free to let me know.
Thank you for inviting me to review this paper entitled “On the playing field to improve: a goal for autism”. The authors aimed to analyze the research results found in literature in terms of improvements and difficulties encountered by the people with ASD who decide to take part in a soccer program.
I have really appreciated the authors’ efforts and acknowledge the novelty and the interest of the topic. However, I have some suggestions that the authors may take into account to improve the quality of their manuscript.
First, even if this is presented as a narrative review, it seems more a hybrid between a narrative and a systematic review. In fact, a search string was used and the results of the search have been reported in a systematic way, with a table and so on. The authors should decide if they want to keep the “narrative” or the “systematic” modality. In this second case, it would be better to implement the Methods section by adding more information of interest for the authors, such as the inclusion criteria for the studies (Participants, Intervention, Outcome, Study design), the timeframe of the included studies, the methodology about data extraction of results report, and so on….
In case you decide to keep it as a narrative review, I would suggest giving some critical interpretation of the results, instead of simply describe the study that deals with a topic or another. For instance, in the paragraph about “The neuropsychological benefits of soccer”, the authors simply report the neuropsychological effects of football in players, but do not explain the rationale for soccer being beneficial for ASD. The same for the other paragraphs. This is important to lead the reader to the “core” paragraph of the review (paragraph 6).
Many thanks for your suggestions. I made more explicit some conclusions about the potential role of soccer for people with ASD. Moreover, I added more critical interpretations of the results of the studies analyzed as you requested (please see lines: 181-183; 208-211; 280-294; 316-320)
Also, in the Table, I suggest to insert more specific information, e.g. mean age/range, sex of participants, type of treatment, treatment duration, comparison (if there is any), outcome (not improvements…), outcome measures (if there are validated scales). This would make the Table clearer for the reader.
Thanks for the suggestions. Unfortunately, not all demographic details (age mean +/- standard deviation and range) are available in the four studies analyzed. The type of soccer program and its duration are summarized in the columns “soccer program” and “duration” respectively, more details are available in the text. A comparison is present only for the study by Barak et al 2019, we specified that in the text (please see lines 263-265). We added an additional column, in table 1, entitled “Outcome measures” in which we summarized the main instruments used to assess soccer skills. In my opinion, thanks to your suggestions, the table 1 is now more complete and clearer.
In general, I suggest the authors be more critical (in sense of data interpretation) and gives also a suggestion which might be useful for clinical practice/rehabilitative programs.
Thanks for the suggestion. We added more critical interpretations despite the limited data and the heterogeneity of soccer programs (please see lines 280-292; 314-318)
Minor comments:
- Line 57-59: “ASD includes a constellation of language, personality, social-behavioral phenotypic traits 58 combined with several frequent neurodevelopmental, medical and psychiatric comorbidities that 59 further complicate the complex picture of autism manifestations.”: Please, delete. It is a repetition of a previous sentence.
I deleted the sentence in accordance to your suggestion.
- Page 2: I suggest to move the aim of the study to the end of the Introduction, instead of merging it with the Methods section, which should be improved, as reported above.
I better explained the aim of the study in the end of the introduction (see lines 88-90).
- Line 109: [3].3 --> [33].
I corrected it.